# Two *Ascophyllum nodosum* Fucoidans with Different Molecular Weights Inhibit Inflammation via Blocking of TLR/NF-κB Signaling Pathway Discriminately

**DOI:** 10.3390/foods11152381

**Published:** 2022-08-08

**Authors:** Lilong Wang, Linlin Wang, Chunhong Yan, Chunqing Ai, Chengrong Wen, Xiaoming Guo, Shuang Song

**Affiliations:** 1National Engineering Research Center of Seafood, School of Food Science and Technology, Dalian Polytechnic University, Dalian 116034, China; 2Shenzhen Key Laboratory of Food Nutrition and Health, Institute for Advanced Study, Shenzhen University, Shenzhen 518060, China

**Keywords:** sulfated polysaccharides, chemical structure, anti-inflammation, molecular weight

## Abstract

The present study aimed to clarify the potential mechanism of fucoidans found in *Ascophyllum nodosum* on anti-inflammation and to further explore the relationship between their structures and anti-inflammation. Two novel fucoidans named ANP-6 and ANP-7 and found in *A. nodosum*, were separated and purified and their structures were elucidated by HPGPC, HPLC, GC-MS, FT-IR, NMR, and by the Congo red test. They both possessed a backbone constructed of →2)-α-L-Fuc*p*4*S*-(1→, →3)-α-L-Fuc*p*2*S*4*S*-(1→, →6)-β-D-Gal*p*-(1→, and →3,6)-β-D-Gal*p*4*S*-(1→ with branches of →2)-α-L-Fuc*p*4*S*-(1→ and →3)-β-D-Gal*p*-(1→. Moreover, ANP-6 and ANP-7 could prevent the inflammation of the LPS-stimulated macrophages by suppressing the NO production and by regulating the expressions of iNOS, COX-2, TNF-α, IL-1β, IL-6, and IL-10. Their inhibitory effects on the TLR-2 and TLR-4 levels suggest that they inhibit the inflammation process via the blocking of the TLR/NF-κB signal transduction. In addition, ANP-6, with a molecular weight (63.2 kDa), exhibited stronger anti-inflammatory capabilities than ANP-7 (124.5 kDa), thereby indicating that the molecular weight has an influence on the anti-inflammatory effects of fucoidans.

## 1. Introduction

As a host immune system defense mechanism, inflammation restores cells to their normal condition after the pathogen invades the tissue structure and its function [1]. Toll-like receptors (TLRs) are conserved groups of host transmembrane recognition receptors that are found in viral and bacterial products, as well as in other pathogens [2]. Lipopolysaccharides (LPS) and gram-negative bacterial endotoxins are mainly sensed by Toll-like receptors-4 (TLR-4). Many researchers have reported that LPS are associated with the nuclear factor kappa-B (NF-κB) and by activating the TLR-4, they promote the excessive production of inflammatory cytokines [3]. TNF-α, IL-1β, and IL-6 are considered inflammation biomarkers, which play critical functions in some chronic inflammatory diseases [1]. Clinically, TLRs, pro-inflammatory cytokines, and cyclooxygenase enzymes are also crucial pharmacological targets for anti-inflammatory drugs [4]. However, most current anti-inflammatory drugs are not entirely operative for chronic inflammation and may have adverse side effects [4]. Hence, more efforts are needed in order to explore anti-inflammatory drugs with no or lower side effects.

As critical biological macromolecules, sulfated polysaccharides are isolated from marine animals, brown algae, and microorganisms [5]. In addition, sulfated polysaccharides have been proven to have anti-inflammatory [1] and anti-oxidation applications [6]. Fucoidan is a water-soluble, sulfate polysaccharide that is found in brown algae, and that mainly contains a large amount of α-(1→3)-linked L-fucose and sulfate groups [7]. Fucoidan from *Saccharina japonica* alleviates the NO production and inflammatory cytokine levels [8]. Furthermore, fucoidan from *Padina commersonii* can inhibit the nitric oxide (NO) production and that of inflammatory cytokines and suppress the inflammatory effects through the down-regulating of the NF-κB pathways [3]. A low molecular weight (8.1 kDa) of the fucoidan from *Laminaria japonica* and consisting of fucose and galactose, is known to alleviate the inflammatory factors in mice lung tissue [9]. These studies shed light on the positive impact of fucoidan in order to prevent inflammation. Given that the anti-inflammatory properties of fucoidans significantly depend on their structures [7], it is essential to analyze the structure of fucoidan found in brown algae. The molecular weight is a kind of structural factor, that can affect fucoidan bioactivities. A previous study demonstrated that a high molecular weight (386.1 kDa) of the sulfated polysaccharides from *Sargassum cristaefolium* showed a greater NO inhibitory activity [10]. However, the low-molecular-weight (5-30 kDa) of fucoidan from *Macrocystis pyrifera* showed stronger anti-inflammatory properties [11]. The relationship between molecular weight and anti-inflammation is still unclear [7] and more efforts are needed in order to obtain a reliable explanation.

East Asian countries, specifically Korea, China, and Japan, extensively use seaweeds (fucoidan-rich substance) as part of their diet [12]. *Ascophyllum nodosum* is one of the most critically economic brown algae, and it is one of the materials used in alginate production [13,14]. Our previous report showed that fucoidan from *A. nodosum* showed several differences in the structural features and reduced the incidence of antibiotic-induced colitis *in vivo* [14]. However, the relationship between the structure and the anti-inflammatory effects has not been fully elucidated. Therefore, the detailed structural information of two fucoidans isolated from *A. nodosum* was elucidated in the current study as well as their anti-inflammatory effects in the LPS-activated RAW 264.7 cells. In addition, the relative molecular weight and anti-inflammatory effects were also discussed. The present work aims to enlarge the knowledge of the structure-activity of fucoidan and to promote the application of the potential for new functional products using *A. nodosum.*

## 2. Materials and Methods

### 2.1. Materials and Chemicals

The brown seaweed *A. nodosum* was provided by Qingdao Brightmoon Seaweed Group Co., Ltd. (Qingdao, China). The standard monosaccharides were purchased from Sigma Chemical Co. (MO, USA). Papain, pectinase, cellulase, 3-(4,5-dimethylthiazol-2-yl)-2,5-diphenyl tetrazolium bromide (MTT), 1-phenyl-3-methyl-5-pyrazolone (PMP), and LPS were obtained from Shanghai Aladdin Biochemical Technology Co., Ltd. (Shanghai, China). All other chemicals of analytical reagent grade were supplied by Sangon Biotech Co., Ltd. (Shanghai, China).

### 2.2. Fucoidan from the A. nodosum Preparation

The fucoidan was isolated from *A. nodosum* by following the reported method [14,15]. Briefly, the brown algae *A. nodosum* was pulverized and then defatted three times. The residue was extracted with an enzyme mixture containing pectinase, cellulase, and papain in order to obtain the *A. nodosum* polysaccharide (ANP).

The ANP (0.5 g) was loaded into the DEAE-52 cellulose anion-exchange column (4.6 cm × 73 cm), and eluted by a 0, 0.2, 0.4, 0.6, 0.8, 1.0, 1.2, and 1.5 mol/L NaCl solution. The eight separated fractions ANP-1, ANP-2, ANP-3, ANP-4, ANP-5, ANP-6, ANP-7, and ANP-8 were collected and further purified through an ultrafiltration tube (30 kDa and 100 kDa). The polysaccharides were collected, dialyzed (molecular weight cut-off 3.5 × 10^3^ Da), and lyophilized. The yields of ANP-6 and ANP-7 were higher than those of other fractions. Thus, ANP-6 and ANP-7 were used for further analysis.

### 2.3. Chemical Analysis

The contents of uronic acid [16], protein [17], total sugar [18], total polyphenols [19], and sulfate ester [20] were determined as described previously.

The purity and molecular weight distribution of ANP-6 and ANP-7 were examined using a Shimadzu HPLC (Shimadzu Co. Ltd., Kyoto, Japan) system connected with the evaporative light scattering detector (ELSD) 6000 and TSK-gel G4000 column with 0.1 M ammonium acetate at a flow rate of 0.4 mL/min [14]. The temperature and airflow rates in the drift tube of the ELSD were 85 °C and 2.5 L/min, respectively.

### 2.4. Monosaccharide Composition Analysis

The monosaccharide compositions of ANP-6 and ANP-7 were assessed by HPLC (Thermo Fisher Scientific, Basel, Switzerland) [21]. Briefly, ANP-6 and ANP-7 were hydrolyzed at 121 °C for 3 h and derivatized at 70 °C for 0.5 h with PMP. The PMP residues were detected with an HPLC system equipped with a PAD detector at 249 nm. The 20 mM ammonium acetate (83%) and acetonitrile (17%) were used as the mobile phase at 1.0 mL/min.

### 2.5. Fourier-Transform Infrared (FT-IR) Spectroscopic Analysis of ANP-6 and ANP-7

Each sample (1 mg) was dispersed in a KBr powder (100 mg). The spectra used an FT-IR Spectrometer (Perkin Elmer, Waltham, MA, USA) in the range of 4000–400 cm^−1^ at 25 °C.

### 2.6. Congo Red Test of ANP-6 and ANP-7

The conformational structures of ANP-6 and ANP-7 were measured by the Congo red test method [22]. Each 1 mL sample solution (0 as control and 2 mg/mL) was mixed with 1 mL Congo red solution. Next, the NaOH solution (1 M) was added in order to obtain the mixtures of different gradient concentrations. Finally, the maximum absorption wavelength of the samples in a range of 400–600 nm was recorded with an ultraviolet-visible spectrophotometer (Perkin Elmer, Waltham, MA, USA).

### 2.7. Desulfation, Methylation, and GC-MS Analysis

The desulfation of ANP-6 and ANP-7 was carried out using the DMSO-methanol method [23]. Next, the desulfation residues were methylated according to the reported method [23,24]. Once the methylated products were hydrolyzed and reduced, the monosaccharides were transformed into the alditol acetates. Following the vortexing, the derivatives were analyzed with a 7980A/5979C GC-MS (Agilent Technologies Inc., Santa Clara, CA, USA) using an HP-5MS column.

### 2.8. NMR Spectroscopy Analysis

The NMR measurements of ANP-6 (35 mg) and ANP-7 (35 mg), once individually dispersed in 500 μL of D_2_O (99.9%), were conducted using a 400 MHz NMR spectrometer (Bruker Corp., Rheinstetten, Germany) at 25 °C.

### 2.9. Cell Culture

The macrophage RAW 264.7 cells were supplied by the Chinese Academy of Sciences (Shanghai, China). The cells were maintained in Roswell Park Memorial Institute (RPMI) 1640 media from Gibco (Grand Island, NY, USA) and contained 10% heat-inactivated FBS, and were maintained at 37 °C and 5% CO_2_.

### 2.10. Cell Viability Assay

The cytotoxicity induced by ANP-6 and ANP-7 was analyzed using the MTT method [8]. The macrophages were seeded to 96-well plates at a density of 1 × 10^4^ cells/well. Following cell growth for 24 h, the cells were cultured with a 100 μL serum-free medium sterilized using a 0.22 μm sterile filter that contained ANP-6 or ANP-7 at different concentrations. Twenty-four hours later, 20 μL MTT of the solution (5 mg/mL) was discarded. Following a further 4 h incubation, 150 μL of DMSO was added in order to dissolve the formazan crystals that had formed and the absorbance was measured at 490 nm.

### 2.11. Nitric Oxide (NO) Production Determination

The NO production was performed following the previous report [8] with minor modifications. The macrophages (1 × 10^4^ cells/well) were plated in 96-well plates and then cultured for 24 h. Next, the different concentrations of ANP-6 (0 to 100 μg/mL) and ANP-7 (0 to 200 μg/mL) in the serum-free medium were added to the different wells for 2 h, and then the LPS was added in order to achieve the final concentration of 1 μg/mL. The LPS (1 μg/mL) was used as a positive control, and an equal volume of the serum-free RPIM 1640 medium was used as a blank control. Once cultured for 24 h, the nitrite levels were measured using a NO assay kit according to the manufacturer’s protocols.

### 2.12. Real-Time PCR Analysis

The suspended cells (6 × 10^4^ cells/well) were added to 12-well plates and cultured for 24 h. The culture media of each well was then removed using a pipettor and the cells were then washed three times with the sterilized PBS. Next, 1 mL Trizol Reagent (Sangon Biotech Co., Ltd., Shanghai, China) was added to each well and the cells were then removed and placed into a 1.5 EP tube (Sterile and DNase/RNase-Free) using a cell lifter and pipettor, according to the manufacturer’s protocols. The total mRNA was reverse-transcribed into cDNA using a Fast Quant RT Kit (Takara Biomedical Technology Co., Ltd., Dalian, China). Furthermore, the quantitative RT-PCR reactions using the cDNA templates were performed using SYBR^®^ Premix Ex Taq™ II (Takara Biomedical Technology Co., Ltd., Dalian, China) on a qTOWER detection system and calculated using the 2^−ΔΔCt^ method. The primers are shown in Appendix A.

### 2.13. Statistical Analysis

The data are presented as the mean ± SD. The Student’s *t*-test or Tukey’s multiple comparisons test was performed for the statistical differences between the groups. The value of *p* less than 0.05 was indicated as extremely significant.

## 3. Results and Discussions

### 3.1. Physicochemical Properties and the Monosaccharide Composition of ANP-6 and ANP-7

The crude fucoidan was isolated from *A. nodosum* through an enzymolysis treatment yielding 1.3% of dried materials. The ANP was classified by anion-exchange chromatography in order to afford the main fractions of ANP-6 and ANP-7 with a yield of 24.61% and 19.23% from the crude fucoidan, respectively (Figure 1A). Next, the purity and the molecular weight of ANP-6 and ANP-7 were determined using HPGPC. The single and symmetrical peaks are shown in Figure 1B, implying that ANP-6 and ANP-7 are homogeneous fucoidan. Furthermore, the average molecular weights of ANP-6 and ANP-7 were 63.2 kDa and 124.5 kDa, respectively. These results were lower than the previously reported fucoidan from *A. nodosum*, namely, 361.4 kDa and 1330.0 kDa [25,26]. The chemical contents are shown in Table 1. Following the analysis using the Student’s t-test, the contents of total sugar, uronic acid, sulfate, protein, and total polyphenols were not significantly different between the ANP-6 and ANP-7 fractions.

Following the acid hydrolysis and derivatization with PMP, the derivatives of the standard monosaccharides, ANP-6, and ANP-7 were analyzed using HPLC-photo-diode array (PDA) detectors at 249 nm. As is shown in Figure 1C, ANP-6 and ANP-7 were mainly composed of Man: Gal: Fuc, within a ratio of 1.0: 7.8: 31.2 and 1.0: 9.8: 49.2, respectively. In previous reports, fucoidan from *A. nodosum* were usually combined with GlcA, Xyl, Glc, and Man [6,22,23]. Thus, the ANP-6 and ANP-7 isolated from *A. nodosum* may have a different structure, compared with fucoidan from the same species.

### 3.2. FT-IR Spectroscopy Analysis of ANP-6 and ANP-7

The structural information of ANP−6 and ANP−7 was further confirmed by the FT-IR spectra recorded in the region of 4000–400 cm^−1^. In the FT-IR spectra (Figure 1D), the peaks at 3454 cm^−1^ and 2945 cm^−1^ were assigned to the stretching vibration of -OH bonds and C-H bonds [27]. The band at 1648 cm^−1^ corresponded to the water scissoring vibration [28]. The intense band at 1043 cm^−1^ was associated with the stretching vibration of the glycosidic linkage C-O [29]. In particular, the signal at 1238 cm^−1^ in ANP-6 and ANP-7 represented the asymmetrical S=O stretching vibration [30]. The peak at 845 cm^−1^ was caused by the stretching vibration of the symmetrical C-O-S in an axial position and that in the band at 592 cm^−1^ was assigned to the S-O stretching vibration [30,31]. Following desulfation, the bands of dS-ANP-6 and dS-ANP-7 at 1238 cm^−1^ and 845 cm^−1^ disappear, indicating that the sulfate groups were successfully removed. The results confirmed that ANP-6 and ANP-7 contained sulfate, which was consistent with the results of the chemical composition.

### 3.3. The Helix-Coil Transition of ANP-6 and ANP-7

The conformational structure of polysaccharides may also be attributed to their bioactivities [22], therefore the helix-coil transition of ANP−6 and ANP−7 were analyzed using the Congo red test. The analysis results for ANP−6 and ANP−7 are illustrated in Figure 1E. With the increased concentrations of NaOH, the maximum absorption wavelength of the Congo red-fucoidan complexes increased rapidly, indicating that ANP−6 and ANP−7 both have a triple-helix conformation [22]. The fucoidan from *Sargassum fusiforme* also displayed the triple helix structure which was damaged by the 0.2 M NaOH solution [32]. However, when the NaOH concentration reached 0.50 M, the triple helix structures of ANP−6 and ANP−7 were not destroyed. Those results suggest that ANP−6 and ANP−7 adopt a highly ordered conformation even under strongly alkaline conditions.

### 3.4. Methylation Analysis of dS-ANP-6 and dS-ANP-7

The existence of sulfate groups in acidic polysaccharides could affect their dissolution in a methylation solvent, so desulfation was used to simplify the methylation analysis [33]. The methylation analysis results of dS-ANP-6 and dS-ANP-7 are shown in Table 2. The primary type of methylated alditol acetates was attributed to 2,4-di-O-methylfucitol in dS-ANP-6 and dS-ANP-7, suggesting that their backbone was mainly composed of →3)-Fuc*p*-(1→ residue. The position information of the sulfate groups and the branch chains were further analyzed and combined with the NMR results.

### 3.5. NMR Spectroscopy Analysis of ANP-6 and ANP-7

The NMR spectroscopy analysis can provide detailed structural information, including sequences, linkage formations, and the type of sugar residues [27]. Five signals mainly appeared in the range of 4.40–5.50 ppm and are observed in Figure 2A,B, suggesting that ANP-6 and ANP-7 have α- and β-configuration glycosidic bonds. Additionally, the anomeric proton of 5.34 ppm, 5.20 ppm, 4.68 ppm, 4.48 ppm, and 4.42 ppm are assigned to H-1 of B, A, C, D, and E. Five signal peaks of 98.5 ppm, 100.1 ppm, 102.7 ppm, 103.3 ppm, and 103.8 ppm correspond to the C-1 residues of B, A, C, D, and E (Figure 2C,D), respectively.

The spin correlated spectroscopy (COSY) experiment reflects the correlation between the adjacent protons of a sugar residue and the heteronuclear single-quantum correlated spectroscopy (HSQC) reflects the correlations between the protons and the adjacent carbons [34]. According to the report’s data of the fucoidan isolated from the seaweed [35,36,37], the cross-peaks in COSY (Figure 3A,B), and the HSQC spectra signals (Figure 3C,D) of ANP-6 and ANP-7 are assigned and summarized in Table 3. The total correlated signals were further assigned based on the interpretation of the TOCSY spectroscopy (Figure 4A,B), which could be assigned as →2)-α-L-Fuc*p*-(1→ (A), →3)-α-L-Fuc*p*-(1→ (B), →3,6)-β-D-Gal*p*-(1→ (C), →6)-β-D-Gal*p*-(1→ (D), and →3)-β-D-Gal*p*-(1→ (E), which were in agreement with the methylation results. The sulfate groups of the fucoidan are usually located at the C-2 or C-4 positions of fucose residues [38]. Compared with previous reports [39], the chemical shifts of C4 signals of residue A significantly varied downfield at 81.0 ppm compared with no sulfated fucose residue with C4 at 71.0–73.0 ppm. The O-sulfate group could affect the adjacent carbon chemical shift (+6–10 ppm) [34]. The FT-IR results were also shown bands around 845 cm^-1^, indicating the majority of sulfates were at the equatorial C-4 positions [36] in ANP-6 and ANP-7. Thus, combining the analysis results of NMR and FT-IR, residue A was deduced to be a →2)-α-L-Fuc*p*4*S*-(1→. Using the same methodology, residue B with C2 at 78.2 ppm and C4 at 80.3 ppm and residue C with C4 at 77.3 ppm were deduced to be →3)-α-L-Fuc*p*2*S*4*S*-(1→ and →3,6)-β-D-Gal*p*4*S*-(1→ residues, respectively, which correspond with the findings of the fucoidan extracted from brown algae [40,41].

In the NOESY spectra (Figure 4C,D), the inter-residual contacts of H1_A_-H2_A_, H1_A_-H6_C_, H1_B_-H3_B_, H1_B_-H3_C_, H1_C_-H3_B_, H1_B_-H6_D_, H1_D_-H6_C_, H3_C_-H1_E_, and H1_C_-H2_A_ were observed, indicating the existence of A-(1→2)-A, A-(1→6)-C, B-(1→3)-B, B-(1→3)-C, C-(1→3)-B, B-(1→6)-D, D-(1→6)-C, C-(3→1)-E, and C-(1→2)-A linkages. Based on the methylation and the NMR spectrometry data, it could be inferred that ANP-6 and ANP-7 own the same repeating fragment and only differ in their polymerization degree. As shown in Figure 5, their structure is constructed with alternating →3)-α-L-Fuc*p*2*S*4*S*-(1→, →2)-α-L-Fuc*p*4*S*-(1→, →6)-β-D-Gal*p*-(1→ and 3,6)-β-D-Gal*p*4*S*-(1→, and branched with →2)-α-L-Fuc*p*4*S*-(1→ and →3)-β-D-Gal*p*-(1→ residues. These structural results are different from the reported sulfated polysaccharide from *A. nodosum*, which has a backbone of →2)-α-D-Glc*p*A-(1→2)-α-D-Glc*p*A-(1→6)-α-D-Gal*p*-(1→2)-α-D-Glc*p*A-(1→ with a branch chain of T-α-D-Glc*p*-(1→4)-β-D-Xyl*p*-(→3)-α-L-Fuc*p*4*S*-(1→ at the O-3 of 3,6)-α-D-Gal*p*-(1→ residue [42]. Some researchers recently reported that the fucoidan isolated from *A. nodosum* has a backbone of →3)-α-L-Fuc*p*2*S*-(1→4)-α-L-Fuc*p*2*S*3*S*-(1→ [36,43], which was different in the present study. Those differences may be owing to the different extraction methods [7]. In addition, ANP-6 and ANP-7 had relatively lower molecular weights, which could be an advantage for their applications for anti-inflammation [7].

### 3.6. Effects of ANP-6 and ANP-7 on Cell Viability and NO Production

Prior to assessing the anti-inflammatory properties of ANP-6 and ANP-7, the toxic impact of ANP-6 and ANP-7 was checked using the MTT viability assay. As shown in Figure 6A,B, both ANP-6 and ANP-7 exhibited no toxic effect on RAW264.7 cells at concentrations up to 100 μg/mL and 200 μg/mL, respectively. On the basis of these results, ANP-6 (0–100 μg/mL) and ANP-7 (0–200 μg/mL) were selected for the following analysis.

The NO production of the RAW 264.7 cells is one of the critical mediators for assessing the anti-inflammatory activity because NO, as a vital participant in inflammatory reactions, could induce pathological complications throughout the progress of cancer [44]. Thus, the effects of ANP-6 and ANP-7 on the NO inhibition of RAW 264.7 cells were analyzed. As shown in Figure 6C, LPS remarkably enhanced the NO production. However, the NO levels were dose-dependently reversed by ANP-6 (IC50 = 41.44 μg/mL) and ANP-7 (IC50 = 44.66 μg/mL), which were much more effective than the unpurified sulfated polysaccharide from *A. nodosum* [45]. A previous study suggested that polysaccharides with triple-helix configurations showed a stronger inhibitory effect on NO production [46]. Furthermore, both ANP-6 and ANP-7 could reduce the LPS-induced toxicity (Figure 6D), similar to the fucoidan from *P. commersonii* [3]. Therefore, ANP-6 and ANP-7 exhibited excellent inhibitory effects on NO production with no toxicity.

### 3.7. Effects of ANP-6 and ANP-7 on iNOS and COX-2 mRNA Expressions

The production of NO was associated with the levels of iNOS, and the high production of iNOS was often accompanied by upregulated COX-2 in the inflammatory process [3]. The gene transcription properties of iNOS and COX-2 were significantly secreted in the LPS group (Figure 7A,B). However, the up-regulated expression of iNOS and COX-2 could be dose-dependently reversed by ANP-6 and ANP-7. Pretreatment with 100 μg/mL ANP-6 and ANP-7 could decrease the expression of iNOS by 72.22% and 54.17%, respectively. Furthermore, the COX-2 production was significantly reduced by 74.84% and 46.93%, respectively. These results were similar to the fucoidan from *S. japonica* which significantly decreased the mRNA expressions of iNOS and COX-2 in the LPS-induced cells model [44].

### 3.8. Effects of ANP-6 and ANP-7 on the Inflammatory Cytokine mRNA Expression

The abundance of inflammatory cytokines (TNF-α, IL-1β, and IL-6) played a remarkable role in the process of inflammation [1,14]. As illustrated in Figure 7C–E, the mRNA transcription properties of TNF-α, IL-1β, and IL-6 were increased by 2.85-fold, 8.15-fold, and 6.85-fold by the LPS stimulation, respectively. Moreover, ANP-6 and ANP-7 could significantly decrease the TNF-α, IL-1β, and IL-6 levels in a dose-dependent manner. Herein, the IL-6 expression was inhibited to 19.8% by ANP-6 and 37.08% by ANP-7 at 100 μg/mL. However, no obvious difference was observed in the TNF-α level and the IL-1β level between the ANP-6 treatment and the ANP-7 treatment at 100 μg/mL. As shown in Figure 7F, the ANP-6 and ANP-7 fractions could up-regulate the IL-10 level, a critical factor in preventing inflammation. Further analysis of the ratios of pro/anti-inflammatory cytokines (Figure 7G–I) showed that the proportions of TNF-α/IL-10, IL-1β/IL-10, and IL-6/IL-10 could be significantly decreased by ANP-6 and ANP-7 at 50 μg/mL and 100 μg/mL. The ANP-6 treatment significantly decreased the IL-1β/IL-10 ratio and IL-6/IL-10 ratio compared with the ANP-7 treatment at 100 μg/mL. The fucoidan from other brown seaweed could also decrease the expression of the pro-inflammatory cytokines and their half maximal inhibitory concentration (IC50) of pro-inflammatory cytokines expression was reported to be 200 μg/mL [44] and more than 1 mg/mL [47], which was much higher than that of ANP-6 and ANP-7 of 50 μg/mL and 100 μg/mL, respectively. In addition, the IC50 value of the fucoidan from *P. commersonii* was 50 μg/mL [3], which was consistent with ANP-6. These results suggested that ANP-6 showed a better reduction of the IL-6 expression and up-regulation of the IL-10 expression, which suggested its superior anti-inflammatory effect.

### 3.9. Effects of ANP-6 and ANP-7 on the TLR-2 and TLR-4 mRNA Expression

Recently, many research studies indicated that LPS activated the inflammatory response by the TLR-2 or TLR-4 mediated NF-κB signal transduction [1,8]. To reveal the anti-inflammatory mechanism of ANP-6 and ANP-7, the expressions of TLR-2 and TLR-4 were evaluated. The activation with LPS significantly enhanced the expressions of TLR-2 and TLR-4. Both ANP-6 and ANP-7 showed down-regulated gene expressions of TLR-2 and TLR-4 (Figure 7J,K). At a concentration of 100 μg/mL, ANP-6 showed a stronger down-regulation in the expression of TLR-2 and TLR-4, which is consistent with the inhibition of inflammatory cytokines expression.

Previous studies have demonstrated that LPS could induce a macrophage differentiation into the M1-type phenotype by the TLR4-mediated NF-κB signaling pathway, which could lead to the over-production of pro-inflammatory cytokines [1,44]. Some fucoidan extracted from seaweed could protect against the LPS-induced macrophage by inhibiting the TLR/NF-κB signal transduction [1,3,8]. In the present study, the LPS-induced expression increases of TLR-2, TLR-4, TNF-α, IL-1β, iNOS, COX-2, and IL-6 related to the NF-κB pathway, were all down-regulated by the ANP-6 and ANP-7. Therefore, ANP-6 and ANP-7 may regulate the LPS-induced inflammation by blocking the TLR/NF-κB signal transduction and the possible signaling pathway that is proposed in Figure 7L. Some evidence indicates that polysaccharides with a higher fucose content, β-(1→3, 1→6) linkages, and triple-helix configurations show more remarkable anti-inflammatory capabilities [1,44,46]. So, the high fucose contents, →6)-β-D-Gal*p*-(1→ and →3)-β-D-Gal*p*-(1→ and triple-helix configurations of ANP-6 and ANP-7 may contribute to their potential anti-inflammatory activity. It could be noted that ANP-6 showed stronger inhibitory effects than ANP-7 at the same concentration. As observed in the in vivo models, a high-molecular-weight fucoidan could increase the infiltration of inflammatory cells in the tissue, while the fucoidan with low-molecular-weight could alleviate the phenomenon [9,47]. Based on the above results, ANP-6 showed a stronger capability, which could be attributed to its lower molecular weight compared with ANP-7.

## 4. Conclusions

In the present study, two fucoidans, ANP-6 and ANP-7, were obtained from *A. nodosum*. ANP-6 and ANP-7 have a remarkable similarity in the saccharide chain structure but only differ in molecular weight. Their backbones are both constructed with alternating →2)-α-L-Fuc*p*4*S*-(1→, →3)-α-L-Fuc*p*2*S*4*S*-(1→, →6)-β-D-Gal*p*-(1→, and →3,6)-β-D-Gal*p*4*S*-(1→, and branched with →2)-α-L-Fuc*p*4*S*-(1→ and →3)-β-D-Gal*p*-(1→ residues. Furthermore, The Congo red test shows that ANP-6 and ANP-7 both have a triple helix stereo configuration. Both ANP-6 and ANP-7 can regulate NO production and the mRNA expressions of iNOS, COX-2, TNF-α, IL-1β, IL-6, and IL-10 in the LPS-induced RAW 264.7 cells. In addition, they inhibit inflammation by suppressing the TLR-2 and TLR-4 levels, suggesting the anti-inflammatory properties of ANP-6 and ANP-7 are associated with TLR/NF-κB signaling pathway. Of note, ANP-6, with a lower molecular weight, showed stronger anti-inflammatory effects, providing evidence for the important role of the molecular weight in the anti-inflammatory capabilities of the fucoidan. The findings from the current study indicate the potential application of fucoidan from *A. nodosum* as an anti-inflammatory agent.

## Figures and Tables

**Figure 1 foods-11-02381-f001:**
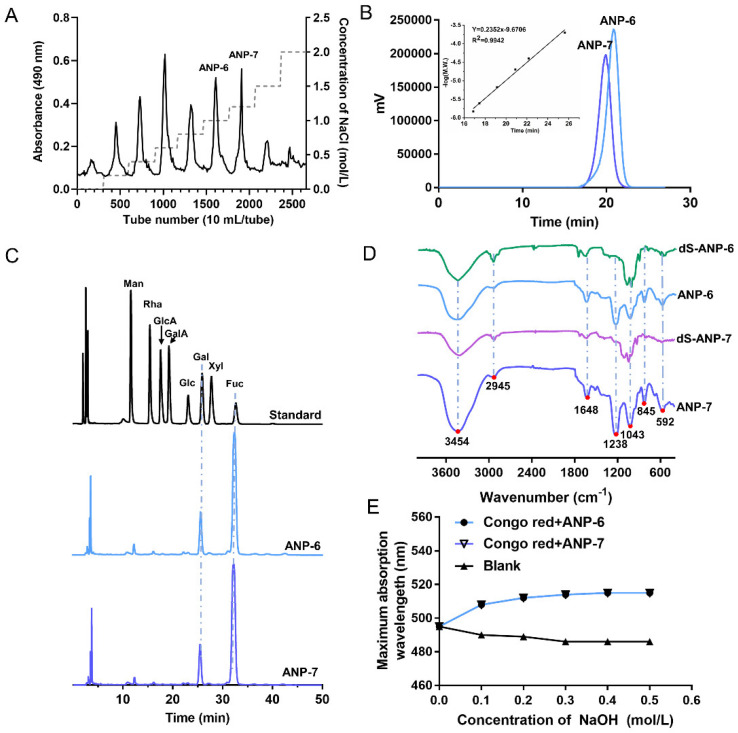
Chemical characteristics of the fucoidan from *A. nodosum*. Stepwise elution curve of the crude ANP on a DEAE−52 column (**A**), HPGPC profiles of ANP-6 and ANP-7 (**B**), FT−IR spectra of ANP-6, dS-ANP-6, ANP-7, and dS-ANP-7 (**C**), HPLC profiles of monosaccharide standards, ANP-6, and ANP-7 (**D**), and the Congo red experiment of ANP-6 and ANP-7 (**E**).

**Figure 2 foods-11-02381-f002:**
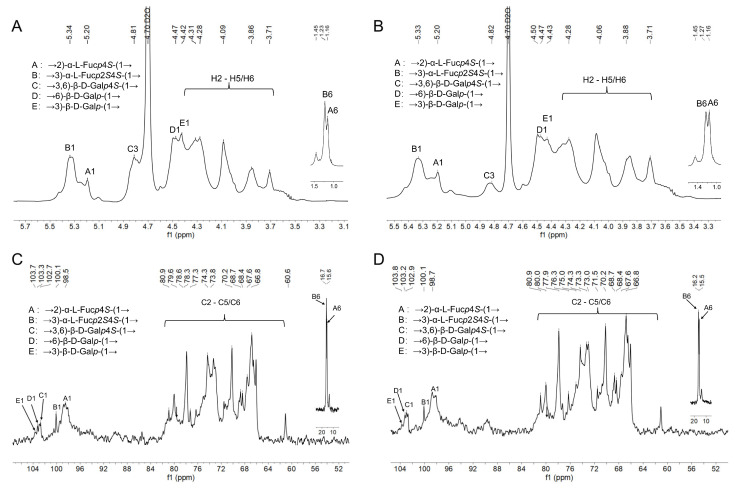
^1^H NMR spectra of ANP-6 (**A**) and ANP-7 (**B**) and ^13^C NMR spectra of ANP-6 (**C**) and ANP-7 (**D**).

**Figure 3 foods-11-02381-f003:**
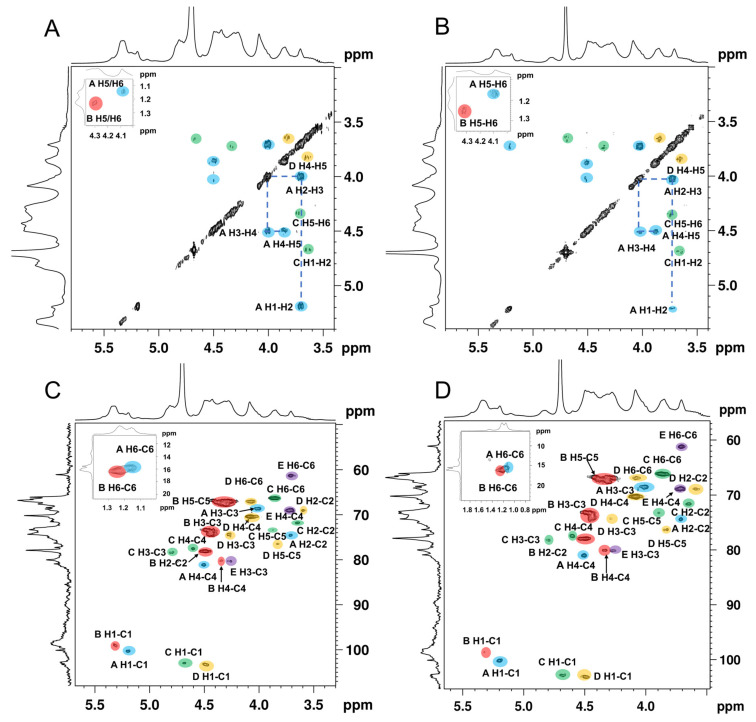
COSY spectra of ANP-6 (**A**) and ANP-7 (**B**) and HSQC spectra of ANP-6 (**C**) and ANP-7 (**D**).

**Figure 4 foods-11-02381-f004:**
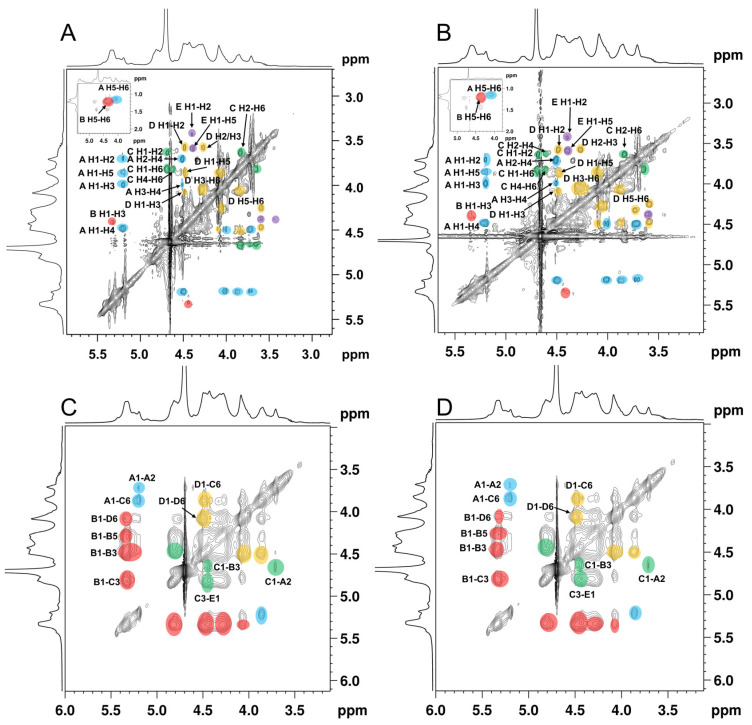
TOCSY spectra of ANP-6 (**A**) and ANP-7 (**B**) and NOESY spectra of ANP-6 (**C**) and ANP-7 (**D**).

**Figure 5 foods-11-02381-f005:**
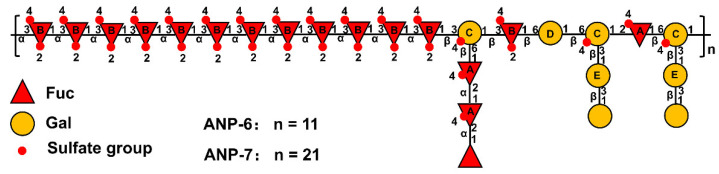
Hypothetical structures of ANP-6 and ANP-7.

**Figure 6 foods-11-02381-f006:**
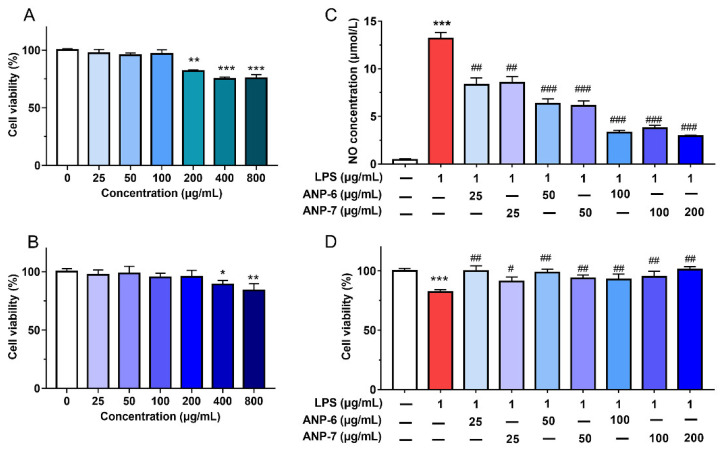
Effects of ANP-6 (**A**) and ANP-7 (**B**) on cell viability and NO production (**C**) and against LPS-induced toxicity (**D**). The results were expressed as means ± SD (*n* = 6). Values are significantly different from the LPS group at ^#^ *p* < 0.05, ^##^ *p* < 0.01, and ^###^ *p* < 0.001. Or * *p* < 0.05, ** *p* < 0.01, and *** *p* < 0.001 against the control group.

**Figure 7 foods-11-02381-f007:**
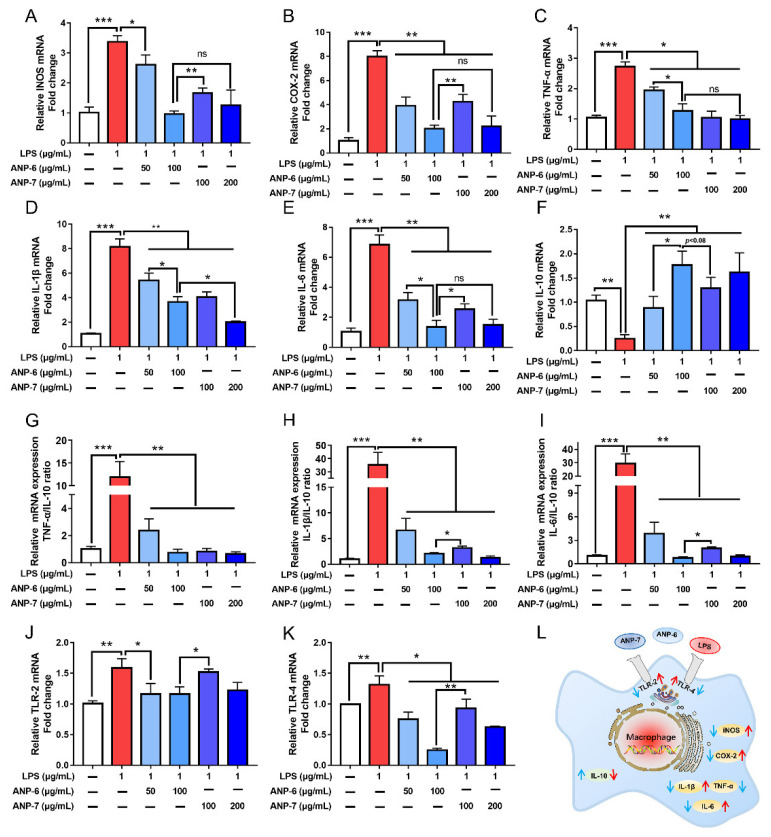
Effects of ANP-6 and ANP-7 on mRNA levels of iNOS (**A**), COX-2 (**B**), TNF-α (**C**), IL-1β (**D**), IL-6 (**E**), IL-10 (**F**), TNF-α/IL-10 ratio (**G**), IL-1β/IL-10 ratio (**H**), IL-6/IL-10 ratio (**I**), TLR-2 (**J**), TLR-4 (**K**) and the possible signaling pathways of ANP-6 and ANP-7 (**L**). Results were indicated as the mean ± SD (*n* = 3). Values are significantly different from the LPS treated or control group at * *p* < 0.05, ** *p* < 0.01, and *** *p* < 0.001. ns, not significant.

**Table 1 foods-11-02381-t001:** Chemical compositions of ANP-6 and ANP-7.

Sample	Total Sugar Content (%)	Uronic Acid Content (%)	Sulfate Content (%)	Protein Content (%)	Total Polyphenols Content (%)
ANP-6	61.67 ± 1.34	3.39 ± 0.29	39.88 ± 1.61	0.01 ± 0.005	0.01 ± 0.002
ANP-7	60.88 ± 1.36	2.70 ± 0.37	42.06 ± 1.48	0.01 ± 0.006	0.01 ± 0.003

Data are expressed as mean ± SD (*n* = 3). Data are the result of a statistical analysis using the Student’s *t*-test.

**Table 2 foods-11-02381-t002:** Methylation analysis and mode of linkage of dS-ANP-6 and dS-ANP-7.

Sample	Methylated Derivative	Deduced Linkage	Characteristic Fragments (*m*/*z*)	Molar Ratio
dS-ANP-6	2,3,4-Me3-Fuc*p*	Fuc*p*-(1→	43, 57, 71, 89, 101, 117, 129, 142, 161, 189	1.1
	2,3,4,6-Me4-Gal*p*	Gal*p*-(1→	43, 57, 71, 87, 101, 117, 129, 143, 161, 205	2.4
	2,4-Me2-Fuc*p*	→3)-Fuc*p*-(1→	43, 58, 85, 89, 101, 117, 131, 159, 173, 189, 201, 233	12.6
	3,4-Me2-Fuc*p*	→2)-Fuc*p*-(1→	43, 87, 99, 129, 143, 159, 189, 201	2.6
	2,4,6-Me3-Gal*p*	→3)-Gal*p*-(1→	43, 71, 87, 99, 101, 117, 129, 161, 173, 233	1.6
	2,3,4-Me3-Gal*p*	→6)-Gal*p*-(1→	43, 71, 87, 99, 101, 117, 129, 143, 161, 173, 189, 233	1.0
	2,4-Me2-Gal*p*	→3,6)-Gal*p*-(1→	43, 58, 87, 99, 101, 117, 129, 142, 161, 189, 201	3.0
	2,4,6-Me3-Man*p*	→3)-Man*p*-(1→	43, 59, 87, 101, 115, 129, 145, 173, 189, 215	0.2
dS-ANP-7	2,3,4-Me3-Fuc*p*	Fuc*p*-(1→	43, 57, 71, 89, 101, 117, 129, 142, 161, 189	1.2
	2,3,4,6-Me4-Gal*p*	Gal*p*-(1→	43, 57, 71, 87, 101, 117, 129, 143, 161, 205	2.3
	2,4-Me2-Fuc*p*	→3)-Fuc*p*-(1→	43, 58, 85, 89, 101, 117, 131, 159, 173, 189, 201, 233	13.6
	3,4-Me2-Fuc*p*	→2)-Fuc*p*-(1→	43, 87, 99, 129, 143, 159, 189, 201	2.4
	2,4,6-Me3-Gal*p*	→3)-Gal*p*-(1→	43,71,87,99,101,117,129,161,173,233	1.6
	2,3,4-Me3-Gal*p*	→6)-Gal*p*-(1→	43, 71, 87, 99, 101, 117, 129, 143, 161, 173, 189, 233	1.0
	2,4-Me2-Gal*p*	→3,6)-Gal*p*-(1→	43, 58, 87, 99, 101, 117, 129, 142, 161, 189, 201	2.9
	2,4,6-Me3-Man*p*	→3)-Man*p*-(1→	43, 59, 87, 101, 115, 129, 145, 173, 189, 215	0.2

**Table 3 foods-11-02381-t003:** Chemical shift assignments of ANP-6 and ANP-7.

Sample	Sugar Residue		Chemical Shift (ppm)					
			H1/C1	H2/C2	H3/C3	H4/C4	H5/C5	H6/C6
ANP-6	A	→2)-α-L-Fuc*p*4*S*-(1→	5.20/100.1	3.71/74.6	4.02/68.9	4.50/81.0	3.86/67.9	1.16/15.6
	B	→3)-α-L-Fuc*p*2*S*4*S*-(1→	5.34/98.9	4.50/78.2	4.42/74.8	4.30/80.3	4.28/67.5	1.23/16.3
	C	→3,6)-β-D-Gal*p*4*S*-(1→	4.68/102.7	3.64/71.4	4.81/78.5	4.60/77.3	3.90/73.7	3.84/66.8
	D	→6)-β-D-Gal*p*-(1→	4.47/103.3	3.58/69.4	4.26/74.5	4.07/70.1	3.86/76.2	4.08/67.1
	E	→3)-β-D-Gal*p*-(1→	4.42/103.9	3.42/71.2	4.25/80.3	3.69/69.1	3.62/73.3	3.65/61.1
ANP-7	A	→2)-α-L-Fuc*p*4*S*-(1→	5.20/100.1	3.71/74.5	4.02/68.9	4.50/80.9	3.86/67.9	1.16/15.5
	B	→3)-α-L-Fuc*p*2*S*4*S*-(1→	5.33/98.7	4.50/78.3	4.43/75.0	4.31/80.3	4.28/67.5	1.27/16.4
	C	→3,6)-β-D-Gal*p*4*S*-(1→	4.68/102.9	3.64/71.5	4.82/78.3	4.60/77.3	3.89/73.5	3.84/66.8
	D	→6)-β-D-Gal*p*-(1→	4.48/103.2	3.58/69.3	4.28/74.5	4.08/70.1	3.85/76.3	4.07/67.1
	E	→3)-β-D-Gal*p*-(1→	4.40/103.8	3.42/71.2	4.25/80.4	3.70/69.1	3.62/73.2	3.65/61.1

## Data Availability

Data is contained within the article or Appendix A.

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
