# Peer review of "Two Ascophyllum nodosum Fucoidans with Different Molecular Weights Inhibit Inflammation via Blocking of TLR/NF-κB Signaling Pathway Discriminately"

_foods, 2022, doi:10.3390/foods11152381_

Round 1
Reviewer 1 Report
Manuscript foods-1811087
Two Ascophyllum nodosum fucoidans with different molecular 2 weights inhibit inflammation via blocking of TLR/NF-κB signaling pathway discriminately
Reviewer’s comments
This manuscript described the influence of molecular weight in the anti-inflammatory activity of fucoidans extracted from A. nodosus. It clearly demonstrated the importance of a well characterized structure to understand the differential effects observed between polysaccharide even with the highly similar fine chemical structure. The experiments were well defined and the conclusion were based on their result. I recommend the major revisions of this manuscript, which were described below.
ABSTRACT
l. 13: this first sentence have no sense... I think authors want to tallk : This research has the aim to ....
l. 17: please verify the nomenclature between fucans an fucoidans.
INTRODUCTION
There is no information if this seaweed could be used as a food ingredient, or if it is consumed in the natural form, or if this seaweed is commercially explored with food end. As the Foods have food science as scope I suggest to insert some of this type of mention in the Introduction topic
l. 42: You can't start a sentence with "and"
l. 43: I suggest you explore some structural details of this sulfated polysaccharides
l. 48-57: Authors must to deep this subject, they must to discuss what kind of structure they are referring to and what is the molar mass average when they talk about low and high mass.
l. 65: You can't start the sentence in this form (And…)
MATERIALS AND METHODS
l. 98: What were the hydrolysis and derivatization conditions? Please inform it in the text.
l. 116: Vertexing or vortexing?
l. 126: Did you inactivate the serum? It is very important when we are culturing immune cells.
l. 129: How did they authors prepared the polysaccharides? How did they solubilized and sterilized?
l. 137: Why did you use two different cell number? At the cell viability, you used 1 x10e4 for each well. I am wondering since 1.5 x 10e5 is too much cells to put in a well of 96 wells plate. Immortalized cells lack their phenotype when they were growing one under another since they have not contact inhibition
l. 138: It is unclear if you put LPS together with ANP samples or you put it only in a positive control weel. Please clarify
l. 142: How many cells / weel?
l. 143: Did you added LPS together with polysaccharides?
l. 144: Please explain in a little more detail. Did you removed the culture media, how did you removed the cells from wells, how did you lysate the cells?
l. 147: Then. Please remove, rewrite the sentence correctly
l. 151: How many independent experiments did you carry out? How many technical replicates did you put in each independent experiments? In spite to used parametric tests as you related here you must to garantee that your data have normal distribution. Did you carried out some normality test before applies these statistical tests?
l. 155-157: Please remove it. I suggest the authors revise deeply the manuscript to avoid this type of error.
l. 159: Crude fucoidan was called ANP?? Please clarify.
l. 161: The ANP-6 and ANP-7 yields was the main obtained yield between all fraction? These were in relation to crude ANP applied at the column or in relation to crude fucoidans? It seens to be low yields.
l. 164: Please revise the english form in all the text. You can not start the sentence using And…
RESULTS AND DISCUSSION
l. 168: I reccomend to analyse the protein content.
L. 178: I agreed that in ANP6 and ANP7 the structures have lowest GlcA, Xyl and Man than the major brown seaweeds fucoidans but we must to consider that there were around 3% of uronic acids and in the chromatographic profile from HPLC showed a peak which eluted around 5 minutes which should be from aldobiouronic acids formed after total acid hydrolysis. Thus, authors must take care when they declare novel structure. Also, these fucans-types have been previously described in another brown seeaweeds.
l. 195: In the figure this band was described as 1043. Please correct it
What about band at 592 cm-1? Please assigned it.
l. 203: Please cite some reference that you based on this statement. Also, I think it is important to discuss the fact that even with considerable difference in molar masses, both fractions have identical profiles.
l. 209-211: The correct form to define correctly the fine chemical structure of sulfated polysaccharides would transform samples in pyridinium salt to proceed methylation process. You should analyse it together with the current analysys of dessulfated samples to provide information about the sulfate position in the structure. Did the authors tried this strategy? I suggest they carry out it.
l. 224-225: This sentence have no sense, please correct it.
Figure 2: Which ones were each letter symbols? Please provide this information at the subtitle of the figure. Please provide more approximated spectra. I suggest provide H-NMR from 5.5 to 3.5ppm and the region between 1.5 to 1.0 ppm put in insert. In case of 13C-NMR spectra I suggest put region between 110 to 55 ppm and the region between 10 to 20 ppm put in insert. Please put the letters in a higher format.
Figure 3 and 4: These very important 2D-NMR spectra should be higher in the final version of manuscript.
l. 275: I think this type of structure could be call fucan and those that have GlcpA and Galp at the main chain could be name fucoidans. Please separe these two sentences and enhance the quality of english form to make more clearly that you are talking about two groups of polysaccharides previously isolated and characterized from A. nodosus.
l. 276: Why? I guess this type of structure also could be found in another A. nodosus extracts but the other authors only did not isolated and characterized it before.
l. 278: Please cite some reference to affirm this comment. Some high molar mass polysaccharides could showed better application than the lower ones.
l. 304: I suggest to modify this statement because NO production is only one of the inflammatory marker to describe these molecules as excellent anti-inflammatory effect.
l. 334: Please change “inhibit” to “decrease”. In which concentration ANP6 decrease it?
Please change “treatment significantly inhibited…”to “treatment significantly decrease…”
l. 323-338: What about the literature data regarding other fucans and fucoidans effects on macrophages? Authors did not mention any literature to discuss their data, please insert a more deeper discussion here.
l. 361: Please change “showed stronger inhibited effects” to “showed stronger inhibitory effects”
l. 364: “Also” instead “And”
l. 367-368: I suggest to remove this sentence since you already discuss it in the above sentences.
CONCLUSION
l. 370-37: Please use the italic font to refer species name.
l. 378: They did not inhibited the anti-inflammatory effect. They inhibited the inflammatory effect.

Author Response
Thank you for your useful comments and suggestions on our manuscript. We have modified the manuscript accordingly, and detailed corrections are listed in the attachment point by point (while marked as the red font in the manuscript).

Reviewer 2 Report
This research did provide some valuable data. However, there are some concerns that should be addressed.
Concerns:
1.In line 81, the authors should explain why they use these three enzymes (pectinase, cellulase, and papain) to extract crude fucoidan?
2.In line 91, the authors should explain why they choose ANP-6 and ANP-7 for further experiments.
3.In line 110, the authors have mentioned that “range of 400-600 nm was recorded by ultraviolet spectrophotometer”, however 400-600 nm is not belonged to UV region.
4.In 2.10, 2.11, 2.12, the authors should explain have they changed the medium to serum-free medium when treating the cells with LPS and ANP-6 and ANP-7?
5.In Table 1, please label the statistical results.
6.In Table 1, do ANP-6 and ANP-7 have polyphenols content?
7.In Figure 1E, what is blank?
8.The authors should explain how they get the data of Table 3?
9.In Figure 7, only mRNA data are provided, generally, the protein level data are necessary.
10.In Figure 7L, the authors have proposed a signaling pathway, when providing a signaling pathway, using inhibitors or antibodies to further confirm the signaling pathway, is necessary.
Author Response

(The authors gave the same response as above.)

Round 2
Reviewer 2 Report
All the comments have been answered.